# The Special Care Unit for People with Behavioral and Psychological Symptoms of Dementia (SCU- B) in the Context of the Project “RECage-Respectful Caring for Agitated Elderly”: A Qualitative Study

**DOI:** 10.3390/ijerph192416913

**Published:** 2022-12-16

**Authors:** Anna Giulia Guazzarini, Georgia Casanova, Friederike Buchholz, Mahi Kozori, Sara Lavolpe, Bjørn Lichtwarck, Eleni Margioti, Aline Mendes, Marie-Louise Montandon, Ilenia Murasecco, Janne Myhre, Elena Poptsi, Valentina Reda, Dorothea Elisabeth Ulshöfer, Sara Fascendini

**Affiliations:** 1Department of Medicine and Surgery, Section of Gerontology and Geriatrics, University of Perugia, 06123 Perugia, Italy; 2National Institute of Health & Science on Aging, Centre for Socio-Economic Research on Aging INRCA (IRCCS), 60124 Ancona, Italy; 3Instituto de Investigación en Políticas de Bienestar Social (POLIBIENESTAR)—Research Institute on Social Welfare Policy, Universitat de València, 46022 Valencia, Spain; 4Department of Psychiatry, Charité—Universitätsmedizin Berlin, Corporate Member of Freie Universität Berlin and Humboldt—Universität zu Berlin, 12203 Berlin, Germany; 5German Center for Neurodegenerative Diseases (DZNE), 12203 Berlin, Germany; 6Greek Association of Alzheimer’s Disease and Related Disorders (GAADRD), 54643 Thessaloniki, Greece; 7Department of Neurology, Humanitas Gavazzeni, 24125 Bergamo, Italy; 8The Research Centre for Age-Related Functional Decline and Disease, Innlandet Hospital Trust, 2313 Ottestad, Norway; 9National Observatory for Dementia and Alzheimer’s Disease, Health Ministry Aristotelous 17, 10433 Athina, Greece; 10Department of Rehabilitation and Geriatrics, University Hospital of Geneva and Geneva University, 1205 Geneva, Switzerland; 11Department of Psychiatry, Geneva University, 1205 Geneva, Switzerland; 12Laboratory of Psychology, Section of Cognitive and Experimental Psychology, Faculty of Philosophy, School of Psychology, Aristotle University of Thessaloniki (AUTh), 54124 Thessaloniki, Greece; 13Primary Care Department, Dementia Programme, Azienda Unità Sanitaria Locale Di Modena, 41124 Modena, Italy; 14Department of Geriatric Psychiatry, Zentralinstitut Fuer Seelische Gesundheit, 68159 Mannheim, Germany; 15Fondazione Europea di Ricerca Biomedica (FERB Onlus), 24025 Gazzaniga, Italy

**Keywords:** dementia, BPSD, special care unit, psychosocial intervention, qualitative study, social innovation, implementation, replicability, RECage project

## Abstract

Background: Dementia is a priority for global public health. The management of behavioral and psychological symptoms of dementia (BPSD) is one of the highest ongoing challenges and needs new approaches. The special care unit for people with dementia and BPSD (SCU-B) is viewed in this context as a further medical intervention. Aim: this study aims to explore SCU-B units in order to describe their main characteristics in relation to different implementation contexts, identify the characteristics of their replicability, and examine the social innovation elements promoted by SCU-B units. Method: This qualitative study is based on focus groups (FGs) and interviews involving nine international centers. Five of the centers have a memory clinic unit and SCU-B, compared with six that only have a memory clinic unit. A total number of 18 FGs were held, which altogether involved 164 participants. All data were transcribed verbatim and analyzed by means of a content analysis and a SWOT (strengths, weaknesses, opportunities, and threats) analysis. Results: The qualitative analysis offers a vision of the SCU-B model as an innovative care unit for BPSD, promoting social innovation in the long-term care (LTC) sector. This system mainly targets people with dementia and BPSD and their informal caregivers but encourages collaboration between dementia care stakeholders at the micro and meso levels. Conclusions: Specific characteristics of the country’s LTC systems and the organization of specialized units are determinants for the success of the SCU-B experience. The replicability of the entire SCU-B model was considered low; however, the implementation of single elements composing the SCU-B model may foster innovation. This study provides relevant suggestions on how to implement the SCU-B unit and innovative solutions for dementia care.

## 1. Introduction

As one of the most common chronic degenerative diseases among the elderly, dementia has become a priority for global public health. In Europe, approximately 10 million people had dementia in 2010 and this number is projected to rise to 19 million by 2050 [1]. According to the World Health Organization (WHO) definition, dementia has physical, psychological, social, and economic consequences for people with dementia, their carers, families, and society at large [2]. In this regard, the WHO has launched the “Global Action Plan on the public health response to dementia-2017–2025” [3] to raise global awareness for the implementation of actions to improve the quality of life of people with dementia (PwD), their caregivers, and families. Dementia represents a tremendous social and welfare challenge due to the progressive decline of cognitive functions associated with the disease and the demanding care load for caregivers and families. Furthermore, the financial and social costs of the disease for families and the National Health System should not be underestimated.

The behavioral and psychological symptoms of dementia (BPSD) affect up to 90% of patients with all types of dementia diagnoses over the course of the disease. These include apathy, depression, anxiety, psychosis, agitation, aggression, sleep disturbances, and other problematic behaviors such as aberrant motor behavior, disinhibition, and resistance to therapy [4].

These symptoms cause severe stress to the PwD and caregivers, and frequently serve as the catalyst for early institutionalization [5,6]. Drug treatment for BPSD is typically highly trusted by caregivers since it appears to reduce their care and emotional burden and appears to affect the amount of time spent on care, supervision, and prevention of dangerous events [7]. However, according to clinical research [8,9,10,11], the efficacy of drugs in treating BPSD is only modest and the use of atypical antipsychotics in older adults with dementia is generally not recommended due to the frequent side effects. Therefore, the off-label use of antipsychotic drugs is widespread in clinical practice and is considered effective.

For this reason, the American Psychiatric Association has published practical guidelines on antipsychotics to treat agitation or psychosis in PwD; they recommend reasonable use only in cases where the symptoms are severe and dangerous for the patient and others [8]. Psychosocial interventions, previously defined as “non-pharmacological”, for managing BPSD should therefore consistently be implemented prior to any pharmacological intervention to avoid adverse events associated with antipsychotic drugs. The psychosocial interventions comprise a range of interventions for managing BPSD and can be directed towards people with BPSD, their caregivers, family members, or the environment. Moreover, interventions using rehabilitation and psychoeducational programs can stimulate and improve the cognitive, behavioral, physical, and sensory skills of the person with dementia [12,13].

The Lancet Commission recently recommended multi-strategic psychosocial interventions as the most effective intervention for reducing agitation and neuropsychiatric symptoms for PwD [14]. In this scenario, the special care unit for patients with dementia and BPSD (SCU-B) represents a new approach for addressing a growing population of people with dementia. An SCU-B is a residential medical structure outside a nursing home, in a general hospital, or elsewhere, for example, in a private hospital, where patients with BPSD are temporarily admitted when their challenging behavior is difficult to control at home. The SCU-B’s mission is to improve patient behavior, while also working to facilitate their return home whenever possible [15].

An SCU-B must be carefully differentiated from the much more common special care unit (SCU). SCU-Bs are specialized units in long-term care facilities developed to provide specialized care for individuals living with dementia and are (1) medical institutions and (2) focused on the needs of PwD and severe BPSD [1,15].

Evidence on SCU-Bs’ short-term efficacy is not robust, but it is encouraging. Some pioneering papers [16,17] and more recent studies [18,19] have demonstrated marked improvements in BPSD during a short stay in an SCU-B. In France, SCU-Bs (also known as unités cognitivo-comportementales) [20] have been widely used for more than ten years within the framework of a National Alzheimer Plan. France is still the only country to have made this choice.

The project “RECage-Respectful Caring for Agitated Elderly” has been contributing to research on psychosocial interventions since 2019. RECage is a European multicenter multi-country research project set up to study (through a pragmatic clinical trial), adapt, and upscale the medical intervention called “The Special Care Unit for patients with dementia and BPSD, SCU-B”. The RECage project involved eleven clinical centers with memory clinics and SCU-B facilities in seven European countries (Italy, Switzerland, Norway, France, Germany, and Greece). Health and long-term care (LTC) systems are organized differently in these countries. There are two main kinds of healthcare services in Europe: the social security system-based Bismarck model (adopted in Germany, Belgium, Switzerland, France, and Greece) and the Beveridge model supported by the National Health Service (used in Italy and Norway). Both health systems are funded on universality, solidarity, and equity. The primary differences between the two are to be found in the way the different services are financed and operated [21]. Moreover, the literature identifies four LTC systems based on distributions of formal or informal care and the level of care needs. This study proposes a transversal analysis of experiences from three different care regimes: familiaristic care regimes (Greece and Italy), mixed care regimes (Germany and Switzerland), and universal Nordic care regimes (Norway) [22,23].

The purpose of this qualitative study is to examine SCU-B units to describe their main characteristics in relation to different implementation contexts and identify the characteristics of their replicability. The qualitative investigation will also look at the social innovation elements promoted by SCU-Bs.

## 2. Methods

The comparative qualitative approach based on content analysis and a SWOT (strengths, weaknesses, opportunities, and threats) analysis is the most appropriate to achieve this study’s objectives. A SWOT analysis enables the evaluation of an organization’s competitive position and develops strategic planning based on fact-based analysis, fresh perspectives, and new ideas [24].

The literature underlines how focus groups and expert interviews are the main SWOT analysis methods [25]. For these reasons, the qualitative study used these two methods involving professionals and stakeholders. Moreover, the chosen methods are suitable for exploring the elements of social innovation (SI).

The European Commission defined SI in 2013 and included it in its policy agenda. SI refers to “any new idea—including products, services, and models—that simultaneously meet social needs—more effectively than alternatives—and creates new social relationships or collaborations, i.e., it is both good for society and enhances society’s capacity to act” [26].

The recent literature offers a conceptual framework of the SI concept applied to LTC, identifying four different areas to promote social innovation in LTC: (a) new policies or revised policies to better meet social and LTC needs; (b) openness of the beneficiary’s target in particular to informal carers; (c) support beneficiaries’ quality of life (QoL); (d) promote collaboration between stakeholders and services [23,27,28].

A central research team composed of a group of experts from within the consortium and an external expert in practical qualitative studies designed the research framework and prepared the investigation tools. The qualitative study was conducted in eleven clinical centers. Five of these clinics have a memory clinic unit and SCU-B: Gazzaniga (Italy), Modena (Italy), Geneva (Switzerland), Ottestad (Norway), and Mannheim (Germany). The remaining six organizations only have a memory clinic unit: Bergamo (Italy), Mantova (Italy), Perugia (Italy), Berlin (Germany), and Athens and Thessaloniki (Greece).

### 2.1. Procedures

The qualitative study was conducted locally by research teams in their own country, supported by methodology experts. The research comprised two steps: (a) the redaction of a country form; (b) the implementation of a qualitative study through interviews and focus groups.

#### 2.1.1. Redaction of Country Form

All participating centers filled out a detailed form concerning each country’s sociopolitical context and regulatory framework in order to clarify the LTC and health contexts in the countries involved. The contextual data module also gathered the SCU-B’s technical requirements if the unit was operational. This form was mailed to each center’s principal investigator, who then provided a written response.

#### 2.1.2. The Qualitative Study’s Implementation

Detailed methodological guidelines were drawn up to ensure homogeneity in data collection. These covered: (a) the study’s objectives; (b) the sample’s selection criteria; (c) questions on the main topics for the interview and focus group; (d) methodological suggestions based on Krueger. [29,30,31].

The following were the main topics for the FGs or interviews:A SWOT analysis to highlight the SCU-B’s strengths, weaknesses, opportunities, and threats;Identification of the social innovations (SI) promoted by the SCU-B;Pinpointing the scaling-up characteristics of the SCU-Bs.

FGs working in centers that did not have an SCU-B, and therefore, lacked any firsthand experience, were required to answer a series of questions about SCU-Bs.

### 2.2. Participants

The methodological guidelines recommended that local teams separate professionals and stakeholders, thus creating two specialist focus groups. Unit directors were involved mainly through interviews. Table 1 depicts the profiles included in the sample. Priority was given to including professionals with SCU-B experience within the sample on SCU-B centers. Participants who were members of staff were recruited by means of an invitation from the manager or the unit’s director. External stakeholders were invited by email.

The total number of FG participants varied from 5 to 12 in each group section. “External” stakeholders were involved in eight of them. As shown in the following Table 2, a total of 18 FGs were held, which altogether involved 164 participants. The number of expert and stakeholder interviews (either in-person or over the phone) was 22.

FGs and interviews were conducted by the centers from 2019 to 2021. COVID-19 pandemic restrictions caused some delays and necessitated certain changes to the methodology. Some local teams decided to hold FGs by means of videoconferencing and online interviews (Berlin and Modena) or to only gather data from FGs (Greek partners). In five centers, the study was conducted using interviews with experts and FGs (Gazzaniga, Mantova, Bergamo, Perugia, and Mannheim). However, the availability of local stakeholders in certain countries was strongly influenced by the lockdowns or workloads due to the COVID-19 emergency. Two FGs used a mixed strategy that simultaneously involved experts and stakeholders (Perugia and Bergamo). In Modena, three FGs were held involving stakeholders and professionals, as well as multiple interviews with two caregivers that explored the user’s/caregiver’s perspective.

### 2.3. Data Analysis

The interviews and focus group discussions were recorded and transcribed verbatim. Country reports that were translated into English were made possible through the thematic and content analysis of transcriptions. National reports were subjected to a content analysis by the authors, who also post-categorized the data [32,33]. Summaries of the results were produced as a result of comparing the content analysis of general data forms and country reports.

In order to comply with interviewing privacy regulations and to maintain anonymity, participants’ comments were listed using abbreviations linked to their regions (G—Gazzaniga; Ma—Mantova; Ge—Geneva; O—Ottestad; M—Mannheim; Be—Berlin; Mo—Modena; B—Bergamo; P—Perugia; A—Athens; T—Thessaloniki) and the user data collection channel (FG—focus group; I—interview).

## 3. Results

### 3.1. The LTC Context and Specialized Services: Results from Context Forms

The centers participating in the study were integrated into the health and LTC context with significant differences. The analysis of data collected through context forms enabled us to gather relevant suggestions on three main themes potentially influencing the implementation of the SCU-B: (a) LTC strategy, (b) residential care, and (c) memory clinic.

LTC strategy: General practitioners are universally recognized as the focal point of the evaluation and care path. Second, in many countries, dependent older people receive social care. For example, in Germany, social care for people with dementia is graduated by the level of dementia and is specifically designed to cover the related care need [34]. Moreover, dementia care in central and Nordic countries is managed as a social issue and local institutions (such as municipalities) provide support services for families and caregiver stakeholders.

Residential care: Residential care is a crucial component of the dementia care system in all countries, but it is only in Nordic countries that the provision of dementia care is virtually entirely publicly funded. In Norway, the full public funding of the cost of residential care results in a high percentage of persons with BPSD entering a nursing home. In other countries, families must contribute at least in part to cover the overall cost. For this reason, most families in Greece are unable to afford the high expense of a nursing home. Beneficiaries of partial cost assistance for residential care in Italy are those families with specific socioeconomic traits and care recipients with a high level of dependency.

Memory Clinic: Memory clinics are widespread across the countries participating in the study. Memory clinics grant diagnostic assessment, pharmacological treatment, and follow-up. According to the general rules of cost cover in these countries, access to these clinics may be free or may require co-payment (depending on income and age). For example, there is an active day care center in Thessaloniki (Greece) that functions as both a memory clinic and a day care center and also serves as a training center for family and informal (paid) caregivers.

### 3.2. Technical Requirements for a Model SCU-B

A model of the SCU-B therapeutic approach comprises a combination of psychosocial intervention (such as occupational therapy, physiotherapy, doll therapy, sensory room, etc.) and pharmacological intervention (when needed). A specialized multidisciplinary team provides care in environments that are suitable and welcoming for PwD [15]. Five SCU- B experiences are included in this study. Following Table 3 summarizes the set up of SCU-B involved.

Several features are common to all descriptions:An appropriate, homelike environment: as in Geneva, the SCU-B *“is a place to live, not just a place of care”*. Architectural features, such as a dementia-friendly design, are necessary to create a safe environment [35]. A garden or outdoor area is also recommended.A personalized care approach to caring: There was consensus about the need for a dementia-informed system that should be implemented through culture change and specific training. This is a well-known concept. The adage that “one size does not fit all” is especially true in the case of an SCU-B because the more tailored an intervention is, the more effective it becomes [36].Composition of the SCU-B’s staff: the team should be multidisciplinary involving physicians (geriatricians, neurologists, or psychiatrists), nurses, psychologists, neuropsychologists, speech therapists, physical and occupational therapists, nutritionists, and social workers.Physical restraint policy: Several laws govern the application of physical restraints, which should generally be reduced to a minimum. In fact, by general consensus, physical restraints should be eliminated in elderly dementia patients since they are very likely to cause acute functional decline, incontinence, pressure ulcers, and regressive behaviors over a short period of time [37]. The SCU-B is a place where staff can explore alternative solutions for patients’ safety. Identifying the unmet needs of patients with dementia is crucial since these demands may trigger behavioral problems. This topic is closely linked to the urgent need for individualization of care, focusing on the person’s needs: communication consistency, surveillance, appropriate environments, as well as a flexible team approach based on staff communication and respect for patients’ needs and rights [38].Pharmacological therapy policy: Restraint can also be chemical and is achieved with behavior-modifying drugs, such as tranquilizers and sedatives. The approach involves a mix of psychosocial interventions and drug therapies, favoring the first. In every instance, it is advised to first pursue psychosocial interventions once a behavioral disorder has been identified. Drugs are to be considered if these measures do not work; however, these would depend on the patient’s health issues, the cause of the behavior, and the risk to the patient and to those of others.

As summarized in Table 4, an analysis of the technical requirements of SCU-B experiences makes it possible to determine the elements that characterize an SCU-B unit.

### 3.3. SWOT Analysis for the SCU-B

The main points from interviews and FGs will be schematically summarized using a SWOT matrix with four categories: strengths, weaknesses, opportunities, and threats (Table 5). 

#### 3.3.1. Internal Factors: Strengths

The person-centered approach and personalized style of care are seen to be part of the unit’s culture and are equally established through specific training and the team’s prevailing culture.

*“We use the ‘TIME’ (*The Targeted Interdisciplinary Model for Evaluation and Treatment of Neuropsychiatric Symptoms) [39] *to create a goal for the assessment and treatment of each patient” (FC, O).*

“*We were considering how to better respond to challenging symptoms when we came across the Kitwood approach* [40]*, and we decided to try it out” (I, G).*


*“Working in a large, multidisciplinary team presents a significant advantage in terms of being able to provide in-depth differential diagnoses and person-centered care for PwD” (FG, Be).*


The service’s mission is defined by the ability to respond promptly and flexibly to highly critical situations, avoiding improper hospital admissions.


*“This type of intervention relieves the family*
*and reduces the sense of helplessness of local doctors, geriatricians, or general practitioners” (FG, Mo).*


High pharmacological competence is another strength that enables real pharmacological wash-outs to put an essential drug therapy in place that is better suited to actual needs. Furthermore, the SCU-B model *“offers the possibility of optimizing medical treatment and medication under close surveillance, communicating with caregivers to find common treatment goals, and transferring knowledge and ideas for the period following discharge” (FG, M).*

In addition to drug therapy, psychologists and occupational therapists seek to lessen BPSD by identifying those activities that are the most suitable and appreciated by the patients or, with the help of a nurse or physiotherapist, by restoring functional levels in order to support the person in difficulty.


*“It is necessary to be open-minded and creative and to have strategies other than pharmacology (music or other activities). It is a matter of using all the involved components of effective treatment and tailoring it to the patient. However, this is sometimes impossible because hospitalization at the SOMADEM unit is of a short duration. Nevertheless, when families function well, they can help by describing the behaviors observed at home as well as their own behavior. Patient observation is also critical. There are times when patients are more agitated than others and it is necessary to recognize these moments” (FG, GE).*


The decision about the care approach is strongly correlated to the professionals’ level of motivation.


*“The team is incredibly driven and psychologically invested to support our patients, and they (the patients) can tell the difference” (FG, G).*



*“Can I point out an essential aspect? The staff’s humanity. We never leave the patient or his family alone; we didn’t even do that during lockdown when we were all busy in the Covid wards” (FG, B).*


The Modena experts highlight how “*a solid socio-health network, which shares a global vision of care and health centered on the person, is a great support to the SCU-B experience” (FG, M).*

Beyond the pathology, commitment at the various levels of the network is linked to assessing and satisfying the needs of the individual and his family, as well as producing better and sustainable widespread psychophysical health at home.

The involvement of in-patients’ families is another significant strength of the SCU-B. Families are welcomed in the ward and are invited to participate in activities to acquire strategies suitable for managing their family member’s illness at home.


*“…also, caregivers should receive training on how to cope with these symptoms at home. In addition to training, caregivers should also receive necessary information about dementia so that they know what to expect” (FG, T).*


#### 3.3.2. Internal Factors: Weaknesses

The weaknesses that emerged from the FGs are diverse and are often dependent on the local socioeconomic and sociohealth context. However, there is a common concern about the suitability of the SCU-B’s environment. One such concern is the suboptimal architectural design of some units, which are either not considered to be dementia-friendly or have no access to a secure garden or balcony.


*“The ward space is similar to a standard hospital ward unit, and certain areas are not suited for supporting psychosocial protocols (…), e.g., the doors or the room area, the bed (…); many aspects of the ward are similar to those in other hospitals” (FG, G).*


This point is linked to the weakness that emerged at the SCU-B of Geneva: the mirroring effect, that is, patients exposed to the agitated behaviors of others. This occurrence takes place in everyday interactions and often goes unnoticed by the person engaging in mirroring behaviors and the individual being mirrored. The person unconsciously imitates another’s gesture, way of speaking, or attitude [41,42].


*“Mirroring behaviors can be another risk (…). The patient may be confronted with other very agitated patients or patients with advanced dementia; that kind of situation can be morally difficult for the patient” (FG, GE).*


An element of reflection that emerged in Gazzaniga is the facility’s isolation, both geographically and in relation to the network of local services. Some units, including Modena, complained about the lack of resources, including the lack of a psychologist for family members, the limited number of hours of physiotherapy, and the non-optimal ratio between the number of healthcare professionals and beds.


*“Stakeholders outside of the local area seem to be less aware of the SCU-B unit; the SCU-B’s geographic location makes it difficult to access, and the lack of new public funding prevents improvements to these activities” (I, G).*



*“The general public as well as local practitioners are not so familiar with the memory clinic. Even within the hospital, many professionals do not know what we are doing” (I, MA).*


This aspect is undoubtedly linked to the weaknesses identified in the various FGs concerning the difficulty of providing follow-up visits after discharge, including by telephone, the brevity of the hospitalization period observed by the Geneva center, and the length of the waiting lists for admission found in Gazzaniga.

#### 3.3.3. External Factors: Opportunities

The regions that host the SCU-B seem to be more sensitized to the subject, and natural dementia-friendly communities are being created to foster prevention and social promotion activities. The SCU-B’s integration into the network of services it is placed in also enables other structures to communicate and collaborate more effectively; one example is the outpatient service of a center that links extra-clinical care and a nursery home.


*“The unit is part of a clinic and a research institution and this has fostered a climate of striving for continuous improvement” (FG, M).*



*“Maybe we should be more interdisciplinary in our work. Perhaps what we lack is somatic services. Patients frequently have comorbidities” (FG, O).*


The majority of experts expressed hope for a higher level of involvement from general practitioners, both in the pre-admission phase to improve family and patient compliance during the stay and just before discharge to get the home ready for the patient’s return.


*“In the case of complex patients and difficulty returning home after being discharged, the subsequent assistance project will be organized through reporting to the local unit for multidimensional assessment. Patients return home in all other cases” (FG, MO).*



*“As soon as the patient is hospitalized, it is important to prepare for discharge by thoroughly understanding both the patient’s and caregiver’s circumstances at home. The caregiver must be contacted for this purpose within 24 h of admission” (FG, GE).*


The connection between SCU-B and hospital wards could be strengthened to simplify patient referral procedures, particularly with the geriatrics ward as this is the ward that most supports the SCU-B’s distinctive approach to care.

The involvement of family and caregivers is invaluable for overall patient care. Patient-training activities and psychological support to caregivers improve personal, relational, and environmental dynamics that prevent the onset of BPSD. In addition, these activities ensure that the benefits of hospitalization in the SCU-B are maintained after returning home.


*“It is essential that caregivers become an integral part of the therapeutic process. It is impossible not to include them because, once the patient is discharged, they will be the caregivers, the spouses, the children, etc.” (FG, GE).*


#### 3.3.4. External Factors: Threats

It should be noted that when caregivers attend training courses concurrently with the patient’s hospitalization in the SCU-B, this could make an already stressful situation worse. As emerged from the FG of Perugia, *“the risk may also be that the caregiver’s compliance and the practical effectiveness of the psychoeducational intervention will be reduced” (FG, P).*

Caregivers’ commitment to care prevents them from having free time to dedicate to other family members or to themselves. For this reason, many caregivers decided not to participate in the Modena FG because they perceived the invitation to the SCU-B as an additional task associated with their caregiver role.

A significant threat that was discussed is the high staff turnover, which reduces the time required to ensure that new team members get the supervision and training they need for their work.

Financial pressure and its consequences is often an omnipresent challenge in the healthcare system. For example, experts highlight the discrepancy between the amount of time allotted to each appointment and the actual time that a patient and their family members would need.


*“We sometimes need to respond to a string of emails about pharmacological therapy to alleviate a patient’s situation. However, this approach is not structured; it is based on the individual’s goodwill” (FG, B).*


Last but not least, there is still a stigma associated with residential facilities for the elderly, which are considered to be a “last option”. “*The fragmentary care pathways and the global taking charge of the patient and caregivers also contribute to the stereotype that considers residential structures for the elderly as a last resort” (FG, P).*

The FG participants in Norway also pointed out that the current social discourse does not necessarily support prioritizing older people with severe BPSD when it comes to offering high-quality health services. This issue could be considered as institutional ageism, defined as “laws, rules, social norms, policies, and institutions that unfairly restrict opportunities and systematically disadvantage individuals because of their age” [43].


*“I think the old way of thinking about being old still exists. You are placed in an institution when you do not manage things at home. You are not worth anything anymore. You should just be kept safe, and you are done with your life” (FG, O).*



*“The stigma surrounding dementia and mental illness and its impact on staff need to be actively*
*addressed” (FG,*
*A).*


As acknowledged by the World Health Organization (WHO), the Member States in the Global strategy, and the action plan on “aging and health and through the Decade of Healthy Ageing: 2021–2030”, ageism must be combated on a global scale [44,45].

### 3.4. Potential of the SCU-B’s Social Innovation

In accordance with the SI definitions in the literature, the SCU-B can be regarded as socially innovative insofar as it satisfies a social need that is largely unmet (or only partially met) in participating countries. Additionally, the unit provides patients and their families with strong crisis support from a skilled compassionate team that acts in accordance with the patient-centered approach. The SCU-B is also a privileged place for the training of caregivers under the expertise of healthcare professionals dealing with dementia.

Most participants concurred that the SCU-B does fulfill previously unmet needs of patients and caregivers. There is also a potential to share and expand knowledge about the disease, ways to cope with it, and ways to lessen stigma by fostering communication and exchange regarding the topic.


*“SCU-Bs could play an important role in the dementia care network, as they seem to be a missing part of the puzzle. Currently, there are no similar units in Greece, so SCU-Bs could be a step towards a better quality of life for dementia patients with behavioral and psychological symptoms and their caregivers, especially if these units are easily accessible to everyone and free of charge. It is also possible to reduce the extensive use of SCU-Bs and the abuse of antipsychotic drugs for cases that can be managed non-pharmacologically” (FG, T).*


The distinctive features of social innovation also include considerable dissemination among people. The definition proposed by Howaldt and Schwarz [46] fits into this context: “an innovation can be defined as social to the extent that it is conveyed by the market or by the non-profit sector, it is socially accepted and widely spread in society or some of its sub-areas, adapts to circumstances, and is institutionalized as a new social practice”. Therefore, from this perspective, it is possible to analyze the characteristics of this new intervention method’s replicability in areas where it is not yet present and future implementation is being considered.

### 3.5. The SCU-B’s Replicability

The SCU-B’s replicability was discussed during the data collection process in four Italian centers (Gazzaniga, Mantova, Bergamo, and Perugia) and the German center (Mannheim). However, other centers did not address this issue in their country reports because it was not widely debated during the FGs and did not yield any summarized feedback. As a result, the experts and professionals who had no direct experience in SCU-B units stated that they were unable to answer.

*“Is the SCU-B replicable, and in what way would it be useful?”* was the central question used to elicit responses from experts and professionals.

The Mannheim FG revealed that there is a growing culture in Germany of dementia-friendly hospitals, with an SCU in geriatric clinics and an SCU-B mainly in psychiatric clinics. However, patients often need multidisciplinary care. Therefore, as the best possible solution, the FG suggested a special unit in a psychiatric hospital for PwD, separated from patients with no relevant cognitive impairment, with internal medicine expertise, and easy access to additional medical diagnosis and treatment if needed.

In Italian centers, participants’ discussions provided feedback on the replicability of the SCU-B system, indicating a level of replicability for each element and characteristic composing the SCU-B model. Table 6 summarizes the perceived level of replicability (low, medium, and high) for each SCU-B element and the average level of agreement between Italian centers.

All centers agreed with the possible implementation of interdisciplinary teams. Additional internal elements (for example, periodic follow-up, informal care support group, personalized care, and psychosocial therapy) were rated with a medium-high level of replicability. Case management at home was considered easy to implement by half of the centers and low by the other half. Promoting an active local network involving different stakeholders was considered to be relevant and gauged a medium level of replicability.

In general, the SCU-B model received positive feedback from experts in the different centers, but doubts emerge about the replicability of the complete SCU-B system in other regions.


*“The main reasons are the rigidity of the existing service structure that is not open to new services and units (…). In this regard, some of the participants underline how this unit requires professionals with differentiated profiles, not only for health (…) sometimes it is not easy to find them, also because of the financial restrictions of public health units” (FG, MA).*


Interesting suggestions emerged from the debate in Perugia’s FG. Given the difficulties of setting up a special ward, the participants suggested modifying the two existing Alzheimer’s units according to the SCU-B model. These two units already use a person-centered approach [41], individualized care programs with physical and neuropsychological rehabilitation, and a multidisciplinary team. The lack of resources is the main barrier to implementing the ward unit or complete system, which is deemed to have “low replicability”. Therefore, a financial commitment to putting the SCU-B into practice is essential. However, a redistribution of financial resources risks removing funds from prevention and training programs that are already primarily entrusted to private training bodies, non-governmental organizations (NGOs), or voluntary associations.

## 4. Discussion

The qualitative analysis presents a vision of the SCU-B model as an innovative care unit for BPSD, promoting social innovation in the LTC sector. According to the literature [24,27,28], the study develops an SCU-B model that is designed to respond to social and healthcare needs and promote a personalized care strategy that is centered on individual needs. The SCU-B model also prioritizes the dignity of the patient and caregiver and avoids the possibility of the patient being institutionalized permanently as a result of BPSD [15].

The main strengths emerging from the results are instead linked to three main topics: (a) choosing the general strategy; (b) personal skills and motivation of professionals; (c) specific organization and management characteristics. The application of personalized care and the psychosocial model of care proposed by Kitwood [40] have an impact on the culture of care, bolstering staff motivation and promoting a specific organization of space and care paths. This open approach to the design of care services is characterized by the inclusion of families and informal carers, who are also targeted by the service.

The SCU-B model also supports the integration and coordination of services and stakeholders involved in the care path, promoting a specialized care system rather than the single provision of care. This system primarily targets people with BPSD and their informal caregivers, but also encourages collaboration between dementia care stakeholders at the micro and meso levels. At the micro level, individuals (for example, educators and social workers) who are not normally employed by health units join forces with health and social profiles to collaborate in multidisciplinary SCU-B teams. At the meso level, the SCU-B design recommends the participation of local health institutions and NGOs in a local network of stakeholders to strengthen the global regional response to dementia care. In the long-term, these actions also have the potential to promote cultural changes in the population to counteract the stigma of dementia. The results denote that the stigma of dementia is a social issue affecting many different international contexts [47,48]. However, the specific characteristics of the country’s LTC systems and the organization of specialized units are the determinants of success for the SCU-B unit. In particular, the capacity to introduce innovative solutions is hampered in Mediterranean countries, which are typically characterized by low investments in the LTC and health sectors for specialized residential units. Unsurprisingly, a lack of space and other structural issues in the ward are barriers to more widespread implementation. Due to these factors, participants gave the SCU-B model’s overall applicability a low rating. At the same time, the improvement of existing services through single support services (e.g., informal caregivers support groups or periodic follow-ups) or internal organizational changes (for example, multidisciplinary teams) are seen to be applicable. These results confirm that organizational innovations need a progressive adaptation process in terms of financial structure, managerial resources, and organization. The SCU-B model has the potential to promote social innovation in LTC and dementia care due to its ability to provide integrated services and activities to respond to the unmet needs of people with dementia and their caregivers. Another innovative characteristic is the promotion of internal (between professionals) and external collaboration, made possible by the increase in the number of local stakeholder networks.

## 5. Limitations and Future Developments

The study has some limitations, primarily the low homogeneity of methods used by local teams, which reduced the amount of data collected and the comparability of results. Despite these limitations, this study provides relevant recommendations for the implementation of SCU-B units and innovative solutions for dementia care. Future studies should explore the implementation of the SCU-B unit and other specific services included in the model as drivers of organizational change. A survey on the effectiveness of long-term SCU-B experiences should also be carried out.

## 6. Conclusions

The SCU-B model is an innovative care unit for the systematic care program for dementia (SCPD) that promotes social innovation in dementia care and long-term care. The entire SCU-B model includes different services and activities designed to improve the quality of life of people with dementia and BPSD and their families. The SCU-B model’s high level of complexity reduces its capacity to be replicated as a whole. The comparative results, however, advocate the gradual implementation of services and activities according to the area’s specificities and characteristics and the structures in which an SCU-B unit is to be launched. The facility’s internal culture devoted to promoting personalized care and a multidisciplinary team that provides care in a suitable and friendly environment are the primary elements to take into consideration for new SCU-B designs.

## Figures and Tables

**Table 1 ijerph-19-16913-t001:** Targets and profiles included in the sample.

Target	Profiles
Internal professionals	Physicians (neurologists and geriatricians); psychologists; neurophysiologists; nurses; rehab technicians; educators and occupational therapists.
Local stakeholders	Social and health authorities; non-governmental organizations (for example, the Alzheimer’s Association and family support associations); private care providers; informal caregivers of relatives with dementia.

**Table 2 ijerph-19-16913-t002:** Number of focus groups in each center and number of participants involved.

	Interview	No. Focus Groups	Tot. No. Participants
Gazzaniga (Italy)	3	2	22
Berlin (Germany)		1	5
Geneva (Switzerland)		1	5
Ottestad (Norway)		3	17
Mannheim (Germany)	9	2	26
Mantova (Italy)	4	2	18
Perugia (Italy)	1	1	12
Modena (Italy)	2	4	27
Bergamo (Italy)	3	1	11
Athens (Greece)		1	11
Thessaloniki (Greece)		1	10
Total		18	164

**Table 3 ijerph-19-16913-t003:** Summary of how these SCU-Bs are set up.

Country	Location	SCU-Bs	N° of Beds	Staff
Gazzaniga (Italy): Center of Excellence for Alzheimer’s Disease	In a general public hospital run by a private foundation	2	23 beds each	Geriatricians, neurologists, and psychologists
Modena (Italy): Hospital Unit dementias with High-Intensity Care (NODAIA)	Private hospital	1	25 beds	Geriatricians, neurologists, psychologists, nurses, occupational therapists, and social workers
Geneva (Switzerland): SOMAtic DEMentia unit (SOMADEM)	Specialized geriatric hospital	1	18 beds	Physicians, nurses, psychologists, neuropsychologists, speech therapists, physical and occupational therapists, nutritionists, and social workers
Ottestad (Norway): The Research center for Age-related Functional Decline and Disease	Psychiatric hospital	1	5 beds	Psychiatrist and psychologist
Mannheim (Germany): The Central Institute for Mental Health (CIMH)	Psychiatric hospital with a geropsychiatric department	1	24/22 beds	Multi-professional team

**Table 4 ijerph-19-16913-t004:** Elements characterizing the SCU-B.

Characteristic	Description
Ward special unit	A specialized ward for the treatment of BPSD that also houses the SCU-B independently of the other wards.
Informal caregivers support group (ICSG)	Caregivers are provided the opportunity to share their experiences in support groups. This interaction helps caregivers feel less isolated and frustrated while also providing emotional support and better stress management. Although some peer-led groups do exist, support groups are usually led by professionals.
Follow-up (every six months)	Regular and scheduled check-ups for clinical and pharmacological re-evaluation
Managing cases at home after discharge	Contact and support are provided to caregivers over the phone to help them manage the return home and the BPSD by means of environmental interventions.
Personalized care	The most appropriate and effective treatments are identified based on the characteristics of the patients and their medical conditions.
Psychosocial therapy	Psychosocial interventions refer to different therapeutic techniques, usually classified as non-pharmacological
Rehabilitation therapy	Physiotherapy, speech therapy, and occupational therapy
Multidisciplinary team (MDT)	A group of healthcare workers and social care professionals, who are experts indifferent areas and have different professional backgrounds, are united as a team for the purpose of planning and implementing treatment programs for complex medical conditions.
Active local network	The different health services in the area maintain contact and collaborate for the shared care of the patient and caregivers.

**Table 5 ijerph-19-16913-t005:** SWOT Matrix.

Strengths	Weaknesses
Well-trained, skilled, multidisciplinary team	
Continuous staff education	Non-integrated dementia care network
Person-centered approach	Private healthcare structures
Goal-oriented treatment philosophy	Lack of resources
Psychosocial interventions	Lack of ward space
Possibility to optimize medical treatment	No follow-up visits after discharge
Dementia care network	Concentration of patients with solid needs is too high
No restraint policy	
**Opportunities**	**Threats**
Presence of dementia-friendly communities	Social stigma due to hospitalization in an Alzheimer’s ward and ageism
Possibility of admission directly from the emergency rooms	Stigma related to mental health facilities
Good cooperation with outpatient services and nursing homes	Organizational difficulties within the hospital structure and in the cooperation with other hospitals
Involvement of the caregiver during the stay and preparation for discharge	Stressful job leading to employee turnover
Continuous training of staff and debriefing sessions	Families/caregivers and stakeholders are not sufficiently aware of the service
Presence of several specialized services that may be used by the new collaboration	Unrealistic family expectations

**Table 6 ijerph-19-16913-t006:** Replicability of an SCU-B’s elements.

Elements	Level and Agreement %
Ward special unit	High 100%
Informal caregivers support group (ICSG)	High 75%–low 25%
Follow-up (every six months)	High 75%–low 25%
Managing cases at home after discharge (by phone)	High 75%–low 25%
Personalized care	High 50%–low 50%
Psychosocial therapy (Kitwood)	High 25%–Medium 75%
Rehabilitation therapy	High 25%–Medium 75%
Open multidisciplinary team	High 25%–low 75%
Active local network	Low 75%–Medium 25%–High 25%
All systems	Low 75%–Medium 25%

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
