# Peer review of "The Special Care Unit for People with Behavioral and Psychological Symptoms of Dementia (SCU- B) in the Context of the Project “RECage-Respectful Caring for Agitated Elderly”: A Qualitative Study"

_ijerph, 2022, doi:10.3390/ijerph192416913_

Round 1

Reviewer 1 Report

Dear authors!

Thank you for the opportunity of reviewing this manuscript with aims to explore the SCU-B units to describe their main characteristics related to different implementation contexts and to identify the characteristics of its replicability. I reviewed this manuscript with great interest, but there were some problems for publication.

Overall, the article is not written clearly. However, language-wise, the article requires some editing and proof-reading. There are many instances of grammatical errors throughout the article. For example, line 53 „This system is mainly dressed…“ line 56 „could drive innovation“ line 159 „in most existing“, line 214 „analysis's“ , line 299 and 300 „his“ instead its, line 409 „persons“ instead people. Also, word order in sentences line 499, line 417-418 etc. Finally, the quality of the English should be checked by some native speakers.

The abstract should be reorganized according to the IMRaD model.

Introduction is too extensive and contains a lot of unnecessary facts. I suggest removing some parts especially from lines 76 to 138.

Please shorten the results, that is, remove everything that is not crucially important.

There is a big disproportion of the discussion in relation to the other parts. Please expand the discussion regarding the main results. Likewise, the discussion was a summary of other studies.Thank you for the opportunity of reviewing this manuscript with aims to explore the SCU-B units to describe their main characteristics related to different implementation contexts and to identify the characteristics of its replicability. I reviewed this manuscript with great interest, but there were some problems for publication.

Author Response

Response to Reviewer 1:

Dear reviewer,

Thank you for your review of our paper. We have answered each of your points below.

“Overall, the article is not written clearly. However, language-wise, the article requires some editing and proof-reading. There are many instances of grammatical errors throughout the article. For example, line 53 „This system is mainly dressed…“ line 56 „could drive innovation“ line 159 „in most existing“, line 214 „analysis's“ , line 299 and 300 „his“ instead its, line 409 „persons“ instead people. Also, word order in sentences line 499, line 417-418 etc. Finally, the quality of the English should be checked by some native speakers”

We have corrected the grammatical issues pointed out in the manuscript. Moreover, following your suggestion, the manuscript has been extensively English revisions by a specialized service.

“The abstract should be reorganized according to the IMRaD model”

Done.

“Introduction is too extensive and contains a lot of unnecessary facts. I suggest removing some parts especially from lines 76 to 138” – “Please shorten the results, that is, remove everything that is not crucially important”

The sections' introduction and results have been shortened to keep only the relevant information on our topic.

“There is a big disproportion of the discussion in relation to the other parts. Please expand the discussion regarding the main results. Likewise, the discussion was a summary of other studies.” comment allowed us to reflect on the discussion section and implement it according to your suggestion.

Response to Reviewer 1:

Dear reviewer,

Thank you for your review of our paper. We have answered each of your points below.

“Overall, the article is not written clearly. However, language-wise, the article requires some editing and proof-reading. There are many instances of grammatical errors throughout the article. For example, line 53 „This system is mainly dressed…“ line 56 „could drive innovation“ line 159 „in most existing“, line 214 „analysis's“ , line 299 and 300 „his“ instead its, line 409 „persons“ instead people. Also, word order in sentences line 499, line 417-418 etc. Finally, the quality of the English should be checked by some native speakers”

We have corrected the grammatical issues pointed out in the manuscript. Moreover, following your suggestion, the manuscript has been extensively English revisions by a specialized service.

“The abstract should be reorganized according to the IMRaD model”

Done.

“Introduction is too extensive and contains a lot of unnecessary facts. I suggest removing some parts especially from lines 76 to 138” – “Please shorten the results, that is, remove everything that is not crucially important”

The sections' introduction and results have been shortened to keep only the relevant information on our topic.

“There is a big disproportion of the discussion in relation to the other parts. Please expand the discussion regarding the main results. Likewise, the discussion was a summary of other studies.”

Thank you for your comment allowed us to reflect on the discussion section and implement it according to your suggestion.

Reviewer 2 Report

Dear All,

It was with great interest that I read the paper devoted to a hugely significant area of care of people with Alzheimer's disease. As a reviewer I would like to ask the authors to revise the following:

1)     The title should be rewritten in such a way that no abbreviations are used. Thanks to that,
it will be more comprehensible to a potential reader.

2)     It would be judicious to redraft the Conclusions section. It should not contain references to the aim and methodology of the research. Moreover, the information on the limitations
of the study should be moved to the Discussion section.

The research methodology was selected correctly, taking into consideration the nature of the research. The paper was written in an clear and understandable way, without unnecessary repetitions. The references section contains publications that are up to date.

Author Response

Response to Reviewer 2:

Dear Reviewer,

Thank you for your interest and comments. Our answers to your points are as follows.

“The title should be rewritten in such a way that no abbreviations are used. Thanks to that,
it will be more comprehensible to a potential reader”

Done.

“It would be judicious to redraft the Conclusions section. It should not contain references to the aim and methodology of the research. Moreover, the information on the limitations
of the study should be moved to the Discussion section.”

The conclusions section has been revised according to your suggestion. A separate paragraph has been added for the Limitations and future developments of the study.

the manuscript has been extensively English revisions by a specialized service.

Reviewer 3 Report

Thank you for allowing me to report on this paper. My major concerns are methodological. The article opens with a background to the parent study and reports on that for some time. It then goes on to state it will not be reporting the results of that study but a qualitative study that was being conducted alongside the parent study. The abstract and introduction to the paper should focus more intensely on the qualitative study and not the quantitative study. There are many acronyms used without prior expansion/explanation.

The actual analytical method for this qualitative study or the number of respondents or any coding or thematic framework is not described. It is reported very much like a quantitative paper with some quotes from the respondents (who we do not know). This needs major review for a qualitative paper.

Author Response

Response to Reviewer 3:

Dear reviewer,

Thank you for your careful review. We have considered all your suggestions and answered each of your points below.

“The article opens with a background to the parent study and reports on that for some time. It then goes on to state it will not be reporting the results of that study but a qualitative study that was being conducted alongside the parent study. The abstract and introduction to the paper should focus more intensely on the qualitative study and not the quantitative study."

Following your considerations, we have modified the introduction to be more related to the qualitative study. The abstract has been reorganized according to the IMRaD model. The references to the parent study have also been reduced so as not to confuse the reader as to the main topic of the article.

“There are many acronyms used without prior expansion/explanation.”

We have added an explanation before each acronym.

“The actual analytical method for this qualitative study or the number of respondents or any coding or thematic framework is not described.”

Thanks for the example you linked to the review. We have reorganized our methodological analysis according to this model.

Line 44 – expand ReCAGE for the reader to understand what it means. Done

Line 46 – use of acronym without prior expansion-BPSD, SCU-B). Done

Line 48 – acronym used without explanation - e SCU-B u. Done

Line 51 – capital letters for Focus Groups are not necessary. Done

Line 53 – use of LTC without expansion (Long term conditions). Done

Line 53 – ‘dressed” I do not understand the use of this word here. Perhaps instead of ‘mainly dressed for’, it should read ‘mainly aimed at people with...’ Deleted sentence from the abstract.

Line 55 and 56 is difficult to understand – the statement ‘The direct replicability of the entire model is intensely 55 debated.” By whom is it debated? Where is the evidence for the debate? “However, implementing single elements composing the model could drive innovation. Further comment is quite hard to contextualise this in the abstract when the reader has no idea of the model—changed sentence. The abstract has been reorganized according to the IMRaD model.

Line 69, ‘Dementia’ does not need a capital letter in this instance. Done

Line 73. Add ‘and’ between disease and involves. Done

Line 77- unless Dementia is at the start of the sentence, it doesn’t need a capital D; it is referred to as Dementia otherwise. Done

Line 79-80: “…and other problematic behaviours such as wandering, sexually inappropriate behaviour and refusal of treatment.” Personally, I think some of this statement is pejorative. People with Dementia like to walk, like most people, and there should be safe places they can walk safely. It is only termed ‘wandering’ in a clinical setting where there is no place for people with Dementia to walk safely, and health professionals have no time to walk with them to keep them safe. Similarly, inappropriate sexual behaviours are often when a person with Dementia undresses on the ward (as they believe they are in their own bedroom) or gets into another person bed who they mistakenly believe is their husband or wife. Yes there are other aspects to this but sometimes the ‘sexual’ behaviour is misunderstood.

This is a very good point. We agree with your suggestion and have modified the terminology according to the literature as follows “…and other problematic behaviors such as aberrant motor behavior, disinhibition, and resistance to therapy”. Line 85/86

Line 106, Line 111, Line 130, Dementia = dementia. Done

Line 107 – prolonged QT interval. Delated

Line 136-137 – This quote doesn’t really tell the reader anything except it was recommended, whereas explaining what some if the ‘specific components were, might be helpful for the reader. “Specific multicomponent interventions decrease neuropsychiatric symptoms in people with dementia and are the treatments 137 choice”[21] Done

Line 141 – authors, not Authors. Done

Line 151 for example , not e.g. Done

Line 155: expand SCU. Done

Line 157; what is meant by “are cared for permanently by specially trained staff, inwards” what does ‘inwards mean in this context?

We rephrased the sentence as following: “An SCU-B must be carefully differentiated from the much more common special care unit (SCU). SCU-Bs are specialized units in long-term care facilities developed to provide specialized care for individuals living with Dementia and are 1) medical institutions and 2) focused on the needs of PwD and severe BPSD.” Line 116/119

Line 159-160: a mix of prudent pharmacologic 159 treatment. Should pharmacological treatment not be last resort?

We rephrased the sentence as following: “A model of the SCU-B therapeutic approach comprises a combination of psychosocial intervention (such as occupational therapy, physiotherapy, doll therapy, sensory room, etc.) and pharmacological intervention (when needed)” We also moved this sentence in the section “3.2. Technical requirements for a model SCU-B” Line 276/279. We explain better this concept later in the same paragraph “The SCU-B approach involves a mix of psychosocial interventions and drug therapies, favoring the first. In every instance, it is advised to first pursue psychosocial interventions once a behavioral disorder has been identified. Drugs are to be considered if these measures do not work; however, these would depend on the patient's health issues, the cause of the behavior, and the risk to the patient and to those of others”.

Line 168-169 – if unpublished is it relevant? Delated

Line 169-170 – reference, please. Done

Line 187 “care regimens. Regime is a political coup”, Line 141/142 we preferred not to correct because in the title of the article cited in the references is used “regimes”. Reference: Schulmann K, Leichsenring K, Casanova G, Ciucă V, CorcheÅŸ L, Genta M et al. Social Support and Long-Term Care in EU Care Regimes: Framework conditions and initiatives of social innovation in an active ageing perspective.

Line 203 suggest “In comparison six organizations have a …” rephrased sentence and add in the abstract “Five of the centers have a memory clinic unit and SCU-B, compared to six that only have a memory clinic unit”. Line  172/175 we changed the sentence as following: “The qualitative study was conducted in eleven clinical centers. Five of these clinics have a memory clinic unit and SCU-B”

Line 209 – I am not sure I would call a Swot analysis a qualitative process, it is more quantitative in its nature

Perhaps it would be more correct to say only "comparative approach". In fact, the meaning of the sentence is: we conducted a qualitative study using the strengths, weaknesses, opportunities and threats (SWOT) comparative approach.

Line 224 -225 ; suggest a) new or revised policies… Done

Lines 67 – 71 is one sentence which is difficult to understand because it is so long. Please break down into two or three shorter sentence to enable the reader to follow it. (In this regard, the World Health Organization launches the "Global Action Plan 67 on the public health response to dementia - 2017-2025 [2]," intending to raise awareness 68 in countries to promote actions to improve the quality of life of people with Dementia, 69 their caregivers and families, defining Dementia has physical, psychological, social, and 70 economic consequences for people with Dementia, their careers, families, and society at 71 large [3]) Line 81 to 85 is another very long sentence where clarity is lost.

Lines 110-114 – another very long complex sentence

Line 115 – 119 long complex sentence

Line 122 – 124 – difficult sentence “These interventions can stimulate and enhance the cognitive, behavioral, physical exercise, sensory stimulation skills, or even centered on the care 123 staff, care system, and environment.”.

Line 140 -147 – very long complex sentence; The motivation 140 that prompted the Authors to conceive this project is the challenge posed by the Behavioral and Psychological Symptoms of Dementia (BPSD). RECage is a European multicenter, multi-country Research Project aiming at studying (through a pragmatic clinical trial), 143 adapting, and upscaling a medical intervention "the Special Care Unit for patients with 144 dementia and BPSD, SCU-B," an intervention that, albeit already implemented in some 145 European countries and seeming promising, is not widespread and has not been sufficiently studied so far.

Regarding all “very long complex sentence” or “difficult to understand”, the manuscript has been extensively English revisions by a specialized service.

Line 58 and 59 do not relate to the findings of this study – there needs to be an overarching statement about the findings of this study. The line “The results will be an essential starting point for discussing the final data of the 58 ongoing RECage clinical trial” refer to work that is yet to be completed so is not relevant to this study. Delated

Line 175 -176: The results of the study are expected to 175 be presented in February 2023. So what are you reporting here? Its has not yet been defined or discussed in any detail. Delated

Following your suggestion, the references to the parent study have been reduced so as not to confuse the reader as to the main topic.

Line 393-582. the qualitative data has no number of participants, no time period, no definition of the analytical process, no coding template or themes, no time period for the interviews or the general length/time of the interview period. It has all been written up as a quantitative-based study with a few quotes throughout.

We have also reorganized the entire methodology paragraph.

There are no limitations to the study that are mentioned or examined.

A separate paragraph has been added for the Limitations and future developments of the study.

Round 2

Reviewer 1 Report

Dear authors,

since the corrections made are adequate, I think that the manuscript can be accepted for publication.

Best wishes